# Post-transcriptional regulation of satellite cell quiescence by TTP-mediated mRNA decay

Melissa A Hausburg[1,2†‡], Jason D Doles[1†], Sandra L Clement[1,3], Adam B Cadwallader[1], Monica N Hall[1], Perry J Blackshear[1,4], Jens Lykke-Andersen[1,5], Bradley B Olwin[1]*

[1]Department of Molecular, Cellular, and Developmental Biology, University of Colorado Boulder, Boulder, United States; [2]Trauma Research, Swedish Medical Center, Englewood, United States; [3]Biological Sciences Department, California Polytechnic State University, San Luis Obispo, United States; [4]Laboratory of Signal Transduction, National Institute of Environmental Health Sciences, Durham, United States; [5]Division of Biological Sciences, University of California, San Diego, San Diego, United States

*For correspondence: olwin@colorado.edu

†These authors contributed equally to this work

Present address: ‡Ampio Pharmaceuticals, Inc., Englewood, United States

Competing interests: The authors declare that no competing interests exist.

**Abstract** Skeletal muscle satellite cells in their niche are quiescent and upon muscle injury, exit quiescence, proliferate to repair muscle tissue, and self-renew to replenish the satellite cell population. To understand the mechanisms involved in maintaining satellite cell quiescence, we identified gene transcripts that were differentially expressed during satellite cell activation following muscle injury. Transcripts encoding RNA binding proteins were among the most significantly changed and included the mRNA decay factor Tristetraprolin. Tristetraprolin promotes the decay of MyoD mRNA, which encodes a transcriptional regulator of myogenic commitment, via binding to the MyoD mRNA 3′ untranslated region. Upon satellite cell activation, p38α/β MAPK phosphorylates MAPKAP2 and inactivates Tristetraprolin, stabilizing MyoD mRNA. Satellite cell specific knockdown of Tristetraprolin precociously activates satellite cells in vivo, enabling MyoD accumulation, differentiation and cell fusion into myofibers. Regulation of mRNAs by Tristetraprolin appears to function as one of several critical post-transcriptional regulatory mechanisms controlling satellite cell homeostasis.

## Introduction

Skeletal muscle satellite cells (SCs) are maintained as a quiescent stem cell population (*Schultz, 1976*) possessing minimal cytoplasm, condensed chromatin, and few ribosomes (*Wakayama et al., 1979*). Upon muscle injury, SCs activate and then enter S-phase to generate a proliferating myoblast population that differentiates, repairs skeletal muscle tissue, and self-renews to replenish the stem cell pool (*Cornelison et al., 2001*). Induction of MyoD, a myogenic regulatory transcription factor, is a critical component of this activation cascade. Prior to MyoD induction, and immediately upon satellite cell activation, the p38α/β MAPK pathway is activated, serving as a molecular switch (*Jones et al., 2005*). In a subset of SCs, p38α/β is asymmetrically activated during cell division to promote SC self-renewal and re-acquisition of quiescence (*Troy et al., 2012*). Moreover, aged SCs exhibit an intrinsic self-renewal deficit that arises in part from a failure to asymmetrically activate p38α/β MAPK (*Bernet et al., 2014*). Importantly, post-transcriptional gene regulatory mechanisms, including translational repression, contribute to SC activation and play a key role in the maintenance of the quiescent SC state (*Crist et al., 2012*).

**eLife digest** When muscles are damaged, they can repair themselves to some extent by making new muscle cells. These develop from groups of cells called satellite cells, which are found near the surface of muscle fibers. Once the muscle is injured, the satellite cells are activated and can divide to form two cells with different properties. One remains a satellite cell, while the other forms a 'myoblast' that eventually fuses into a mature muscle fiber. Under normal conditions the satellite cells remain in a dormant state and do not divide, but it is not clear how they maintain this dormant state.

To create a protein, the gene that encodes it is first 'transcribed' to produce a molecule called mRNA, which is then used as a template to build the protein. A protein called Tristetraprolin (TTP) can bind to mRNA molecules and cause them to break down or decay, and so TTP can prevent the mRNA from being used to make a protein.

Hausburg, Doles et al. analyzed satellite cells from uninjured muscle and compared them with those from injured tissue. This revealed that when injured, the satellite cells reduced the abundance of several mRNAs, including TTP. Further investigation found that in satellite cells from uninjured tissue, TTP causes the decay of mRNA molecules that are used to produce a protein called MyoD. As MyoD helps the satellite cells to specialize, this decay therefore prevents the formation of myoblasts and keeps the satellite cells in a dormant state. In contrast, damage to the muscle tissue activates a signaling pathway that ultimately inactivates TTP. This enables more of the MyoD protein to be made and the myoblast population to expand.

When Hausburg, Doles et al. experimentally reduced the levels of TTP inside satellite cells, the cells developed into myoblasts even when the tissue was uninjured. Thus, TTP is an important regulator that allows satellite cells to remain in a dormant state. In dormant adult stem cells, regulation of protein availability by RNA binding proteins, such as TTP, may co-ordinate rapid changes in metabolic state to promptly repair injured tissue. A major challenge will be to identify the group of proteins involved and determine the precise mechanisms involved in regulating their availability.

TTP (Tristetraprolin), a target of the p38α/β MAPK pathway, binds to AU-rich elements within the 3′ untranslated region (3′ UTR) of target transcripts and recruits deadenylases, initiating mRNA decay (*Lai et al., 2000*). Rates of mRNA decay, which fluctuate depending on cellular conditions, are controlled by specialized mRNA binding proteins (RBPs) that bind target mRNAs and activate cellular mRNA decay enzymes (*Parker and Song, 2004*). TTP is the best studied of the four member Tis11 gene family, which is comprised of the genes: *Zfp36* (TTP), *Zfp36l1*, *Zfp36l2* and *Zfp36l3*. In macrophages, TTP suppresses the inflammation response by destabilizing proinflammatory mRNAs until extracellular stimuli activate p38α/β MAPK (*Stumpo et al., 2010*). The p38α/β MAPK signaling pathway then phosphorylates TTP, promoting association with 14-3-3 adaptor proteins and blocking the recruitment of deadenylases, thereby stabilizing transcripts with TTP 3′ UTR binding sites (*Clement et al., 2011*). TTP loss-of-function has profound effects on overall organismal health and is associated with a severe inflammatory syndrome caused by chronic over-expression of Tumor Necrosis Factor (TNF)α mRNA, as loss of TNF receptor function largely rescues TTP KO phenotypes in TTP/TNFR1/2 triple knockout mice (*Carballo et al., 2001*).

Transcripts encoding TTP family members are among the fastest declining and dynamically regulated mRNAs following SC activation, where the transcripts initially decline and then are re-expressed (*Farina et al., 2012*). Consistent with this observation, we identified the MyoD 3′ UTR as a TTP substrate, and show that p38α/β MAPK-mediated phosphorylation of TTP regulates MyoD mRNA decay, thereby regulating SC activation. Therefore our data, combined with recent reports of miRNA-mediated regulation of SC quiescence (*Cheung et al., 2012*; *Crist et al., 2012*), identify post-transcriptional gene regulation as an effective and potent means of maintaining SC homeostasis in adult skeletal muscle.

## Results

### Transcripts encoding RNA-binding proteins decrease in activated SCs

Transitioning from a quiescent SC to an activated SC involves large-scale changes in mRNA expression as well as changes in miRNAs (*Farina et al., 2012*) and subsequent events that permit SCs

to begin DNA synthesis and proliferate as myoblasts (*Jones et al., 2005*). To identify regulators involved in SC activation and myoblast commitment, we compared restricted gene expression profiles of FACS-enriched SCs isolated from uninjured muscle with expression profiles from SCs isolated 12 hr post-injury from wild type and *Syndecan 4* (*Sdc4*)⁻/⁻ mice (*Farina et al., 2012*). Syndecan-4 null SCs are incapable of muscle repair, exhibit severely delayed activation, aberrant MyoD expression, poor growth kinetics, fail to self-renew (*Cornelison et al., 2004*; *Hall et al., 2010*) and thus, transcripts differentially expressed between wildtype and *Sdc4*⁻/⁻ SCs may represent genes involved in SC activation and commitment. Genes unique to wild type SCs that change expression 12 hr post muscle injury but do not change in *Sdc4*⁻/⁻ SC are referred to as the 'WT-S4 gene list' (*Figure 1A*, *Supplemental file 1A,1B*). An unexpectedly high percentage of genes in this list decreased in expression during SC activation (47%), with 'nucleic acid binding: RNA binding,' (GO:0003723; p-value = 8.7e-05) as the most significantly overrepresented gene ontology (GO) term (*Figure 1B*). Among these rapidly down-regulated transcripts are genes encoding mRNA destabilizing proteins and genes implicated in cell cycle entry of quiescent C2C12 cells following methylcellulose-induced G0 arrest (*Sachidanandan et al., 2002*). One of these decay factors, *Zfp36*, which encodes TTP, declines precipitously upon wild type SC activation (*Figure 1C,D*). Additionally, transcripts for two other TTP family members, *Zfp36l1* and *Zfp36l2*, which encode Brf1 and Brf2, respectively, also decrease during SC activation (*Figure 1—figure supplement 1*). Among the minority of transcripts that are up-regulated upon SC activation are genes with known roles in myogenesis, including *Elavl1*, which encodes HuR, a MyoD mRNA-stabilizing protein (*Figure 1E,F*) (*Figueroa et al., 2003*), consistent with the induction of MyoD, which occurs as early as 3 hr following isolation of myofiber-associated SCs (*Cornelison et al., 2004*; *Jones et al., 2005*).

## TTP is present in quiescent SCs and is regulated by p38α/β MAPK upon muscle injury

Unphosphorylated, active TTP protein is short-lived as TTP targets its own mRNA for rapid decay, resulting in low levels of unphosphorylated TTP that are technically challenging to detect (*Brook et al., 2006*). Any skeletal muscle perturbation such as induced injury or immediate dissection following sacrifice, activates SCs and p38α/β MAPK (*Jones et al., 2005*), which in other cell types, phosphorylates and stabilizes TTP (*Tchen et al., 2004*). Mice perfused with paraformaldehyde to fix the tissue in situ prior to muscle dissection revealed Sdc4-marked SCs immunoreactive for TTP (*Figure 2A*). Similarly, rapid dissection of the TA muscle followed by immediate fixation also revealed TTP staining in SCs (*Figure 2B*). Upon scoring for TTP positive SCs, we found no quantitative differences in the numbers of TTP+ SCs when the muscle was pre-fixed by perfusion or rapidly dissected (*Figure 2A,B*). In uninjured muscle, 50% of the Sdc4+ SCs were immunoreactive for TTP and located underneath the basal lamina (*Figure 2A,B,E*). Since steady-state levels of active TTP are very low and difficult to detect due to decay of TTP mRNA by active TTP, the lack of detectable immunoreactivity in all SCs is likely due to TTP protein detection limits as opposed to the absence of TTP (*Brook et al., 2006*).

In contrast to the uninjured state, 1 hr post-injury we found the majority of TTP+/Syndecan-4+ cells were located outside of the basal lamina or encased within the basal lamina (*Figure 2C,E*) with a minority present in the SC niche underneath the basal lamina (*Figure 2D,E*). This observation is consistent with substantial muscle damage, as evidenced by serum IgG permeable necrotic myofibers, observed in injured tissue 30 min following a BaCl₂ induced injury (*Weller et al., 1990*) (*Figure 2F,G*). Upon TTP phosphorylation by p38α/β MAPK, TTP accumulates rapidly as a result of increased TTP message and protein stability arising from TTP inactivation (*Tchen et al., 2004*). Therefore, following an induced muscle injury we expect the majority of TTP to be phosphorylated. Unfortunately, the anti-phospho-TTP antibody proved unreliable for detection of phospho-TTP in muscle sections (not shown) and thus, we queried SCs in injured muscle for a direct target of p38α/β MAPK phosphorylation, MAPKAPK2 (MK2), which phosphorylates TTP (*Brooks and Blackshear, 2013*). 30 min post-injury, 45% of the Syndecan-4+ SCs were phospho-MK2 immunoreactive (*Figure 3A,B*), while uninjured muscle SCs were phospho-MK2 negative (*Figure 2—figure supplement 1A*). Consistent with MK2 activation by p38α/β MAPK, phospho-MK2 was significantly reduced if mice were pre-treated with an intraperitoneal (IP) injection of SB203580, a p38α/β MAPK inhibitor (*Figure 3A,B*).

Dissection of SCs from whole muscle activates p38α/β MAPK (*Jones et al., 2005*) and as expected, SCs are phospho-TTP+ following isolation (*Figure 2—figure supplement 1B,C*;

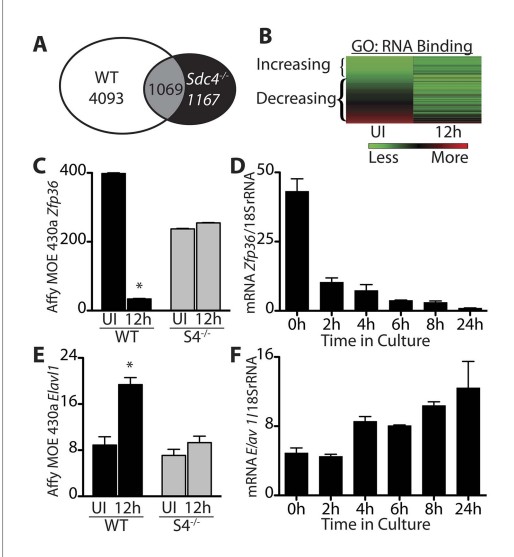

**Figure 1**. Transcripts encoding RNA binding proteins are differentially regulated during SC activation. (**A**) Venn diagram illustrating subtraction strategy used to obtain the WT-S4 gene list. A total of 5162 and 2236 unique genes changed when comparing satellite cells (SCs) isolated from uninjured muscle and muscle 12 hr post-injury in wild type and Syndecan-4 (*Sdc4*) null SCs, respectively. Genes common to both (1069) were subtracted from the wild type gene list yielding 4093 unique genes that changed after injury in wild type but not in *Sdc4*$^{-/-}$ SCs. (**B**) Heat map depicting transcripts filtered from the WT-S4 gene list annotated with the Molecular Function Gene Ontology (GO) term: RNA binding. Transcripts annotated as RNA binding were highly enriched in the WT-S4 gene list (p-value = 8.7e-05) and 70% decreased in wild type SCs but did not change in *Sdc4*$^{-/-}$ SCs 12 hr post injury. Total RNA binding genes: 152; decreased: 107 (70%). (**C**) The relative expression of *Zfp36* decreased in wild type SCs but not in *Sdc4*$^{-/-}$ SCs during the first 12 hr post-injury. (**D**) *Zfp36* transcripts decline rapidly following culture of freshly isolated SCs. (**E**) The relative expression of *Elavl1* increased in wild type SCs but did not change in *Sdc4*$^{-/-}$ SCs during the first 12 hr post-injury. (**F**) Freshly isolated SCs increase *Elavl1* expression upon culture. (**C** and **E** values from SCs isolated by FACS and analyzed on Affymetrix MOE430A gene chips; **D** and **F** determined by QT-PCR where one representative of two experiments is shown, n = 5 mice; * p-value < 0.01. See also **Supplemental file 1A,1B**.

The following figure supplement is available for figure 1:

**Figure supplement 1**. Transcripts encoding the RNA binding proteins Brf1 and Brf2 decrease during satellite cell activation.

*Figure 2—figure supplement 2*). If TTP phosphorylation is mediated via the p38α/β MAPK pathway, then inhibition of p38α/β MAPK should prevent TTP phosphorylation. To test this hypothesis, we injected SB203580 intraperitoneally (IP), isolated SCs in the presence of SB203580, and examined isolated SCs for phospho-TTP to identify activated SCs and examined SCs for HuR induction, which occurs rapidly upon SC activation (*Figure 1—figure supplement 1B,C*). Phospho-TTP and HuR were significantly reduced in SCs isolated from SB203580 treated mice compared to SCs isolated in the absence of the p38α/β MAPK inhibitor, while Syndecan-4 staining remained constant (*Figure 2—figure supplement 1C,D*). Thus, in SCs, TTP appears to be a p38α/β MAPK target and pretreating skeletal muscle with a p38α/β MAPK inhibitor reduces TTP phosphorylation.

## Disruption of the MK2/TTP signaling axis antagonizes MyoD expression

MAPKAPK2 (MK2) is a downstream p38α/β MAPK target regulating TTP in murine macrophage cells (*Mahtani et al., 2001*; *Tiedje et al., 2012*; *Brooks and Blackshear, 2013*). We employed a Duolink in situ proximity ligation assay (PLA) (*Pisconti et al., 2010*; *Troy et al., 2012*) to determine if phospho-MK2 was associated with TTP. As a positive control, we performed PLA using phospho-MK2 and HuR primary antibodies as HuR is a well-established p38/MK2 target (*Kim et al., 2010*). Duolink signals detected in phospho-MK2::HuR samples (*Figure 3C*), were significantly greater than phospho-MK2 PLA using an unrelated RNA binding protein (CUGBP1) antibody where ≤3 complexes per cell were observed (*Figure 3C*) above background. A robust signal for phospho-MK2::TTP association using PLA was observed (*Figure 3C*) and when quantified, was significant compared to contols (*Figure 3D*), indicating that TTP is a bona-fide p38/MK2 target in SCs and MM14 cells.

If TTP is a functional target of MK2, then inhibition of MK2 should reduce or eliminate phospho-TTP expression. Primary myofiber-associated SCs were cultured for 24 hr in the presence or absence of an MK2 inhibitor (MK2 Inhibitor III, CAS: 1186648-22-5) then fixed, stained and scored for Syndecan-4, phospo-TTP, and MyoD. Wild-type, vehicle treated myofiber-associated SCs (Syndecan 4$^+$) uniformly expressed MyoD and were phospho-TTP positive at 24 hr post-isolation, while phospho-TTP was barely detectable in the presence of 10uM MK2 inhibitor III (MK2i, *Figure 3E*). We observed

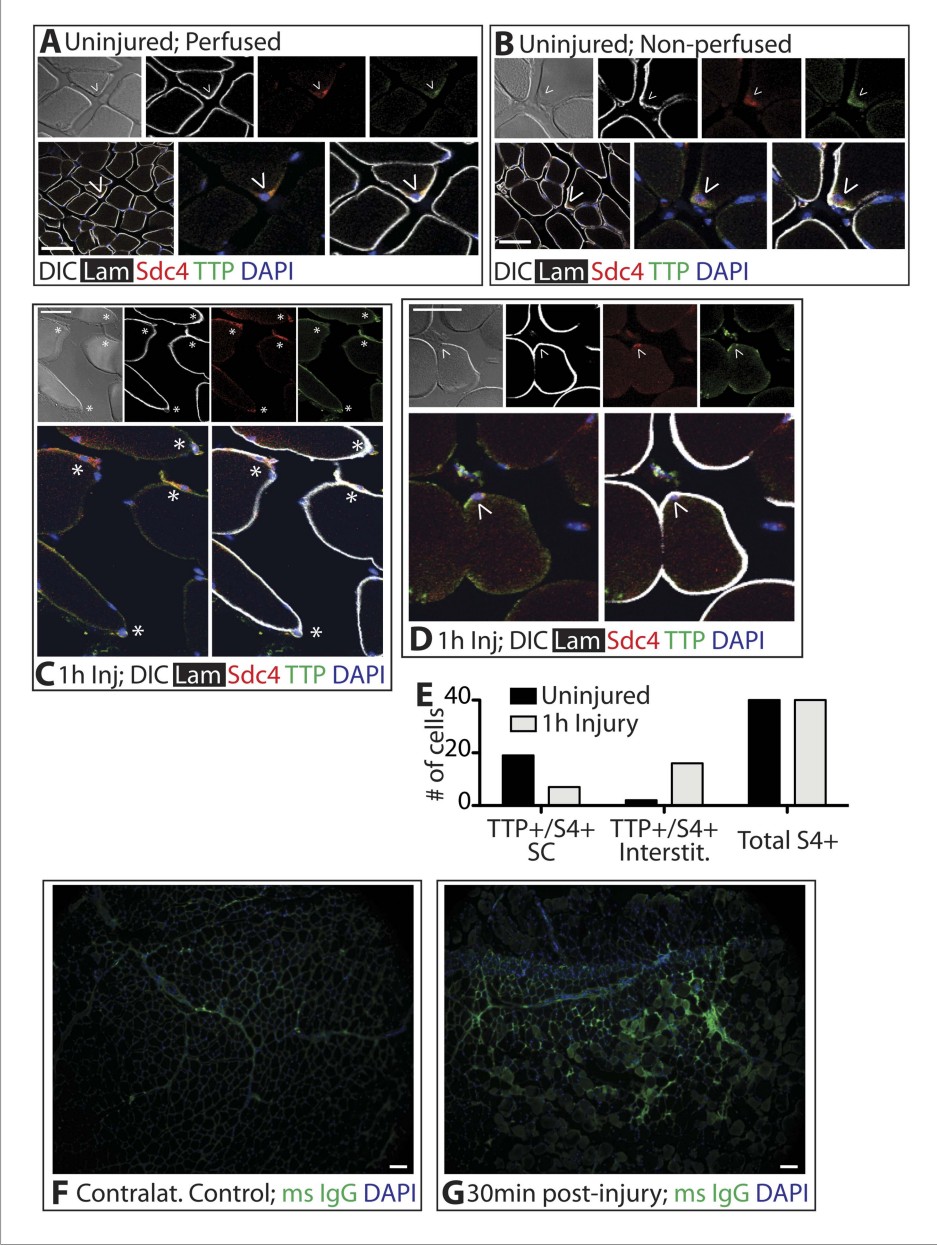

**Figure 2**. TTP positive SCs decrease 1 hr following muscle injury. (**A**) Uninjured perfused TA muscle sections or (**B**) rapidly dissected uninjured muscle sections were stained for Sdc4 (red) and laminin (white) to identify SCs (carets) and for TTP (green). (**C**) and (**D**) TA muscle sections 1 hr following BaCl$_2$ induced injury were stained as in **A** and **B**. (For **A–D**, **F–G**, DAPI is blue; * denotes extralaminar Sdc4+/TTP+ cells). (**E**) The number of Sdc4+ and Sdc4+/TTP+ cells in uninjured TA muscle sections or TA muscle sections 1 hr following BaCl$_2$ induced injury were plotted for two independent experiments. (**F**, **G**) TA muscle sections stained for mouse IgG (green) to assess necrotic fibers in (**F**) uninjured muscle or in (**G**) muscle 30 min following a BaCl$_2$ induced injury. Scale bars = 50 μm (**A–D**), 100 μm (**F**, **G**).

The following figure supplements are available for figure 2:

**Figure supplement 1**. p38α/β MAPK inhibition prevents TTP phosphorylation and reduces HuR protein levels.

**Figure supplement 2**. phospho-TTP expression in myofiber-associated satellite cells.

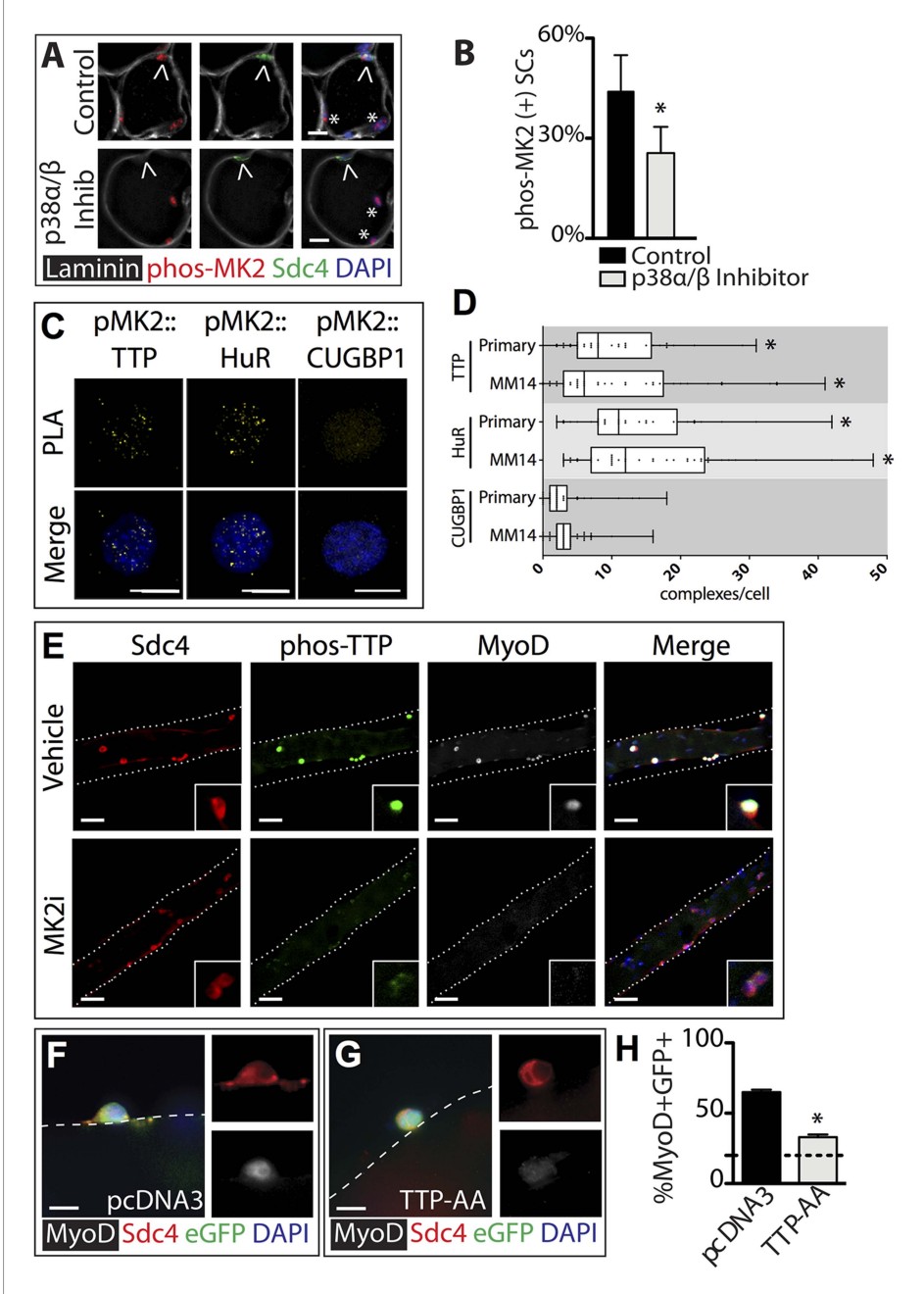

Figure 3. TTP is phosphorylated by p38α/β MAPK and regulates MyoD expression in primary SCs. (A) TA muscle sections harvested 30 min following BaCl₂ injury from either vehicle injected or SB203580 injected mice were stained with anti-Sdc4 antibodies (green) and anti-Laminin antibodies (white) to identify SCs (carets) and anti-phospho-MK2 antibodies (red; * denotes myonuclei positive for p-MK2). (B) Percentage of phospho-MK2 positive SCs was decreased in the presence of SB203580 following injury. (± SEM * p-value < 0.01 for n = 3). (C) PLA in primary, 24 hr cultured SCs using an anti-phospho-MK2 antibody and either an anti-TTP antibody (left column), an anti-HuR antibody (middle column), or an anti-CUGBP1 antibody (right column). (D) Box-and-whisker plots depicting cumulative PLA results from three independent experiments in primary SCs and in the MM14 myoblast cell line (* p-value < 0.01 compared to CUGBP1). (E) Myofiber-associated SCs were cultured in the presence of 10 µM MK2 inhibitor III (MK2i) or a vehicle control (DMSO) for 24 hr, fixed and stained for Sdc4 to mark SCs (red), phospho-TTP (green), and MyoD (white). (F–H) Myofiber-associated SCs transfected 6 hr post isolation with an eGFP plasmid as a transfection marker and either a control plasmid or a plasmid expressing TTP-AA-myc. Explanted SCs were cultured for an additional 24 hr, fixed and stained for Sdc4 (red), eGFP (green), and MyoD (white). (H) The

*Figure 3. continued on next page*

*Figure 3. Continued*

percentage of transfected SCs scored for MyoD plotted for three independent experiments from a minimum 25 myofibers per condition. Scale bars = 10 µm (**A**), 4 µm (**C**), 30 µm (**E**), and 10 µm (**F**, **G**).
The following figure supplement is available for figure 3:

**Figure supplement 1**. TTP over-expression does not affect HuR protein levels.

concurrent downregulation of MyoD in the MK2i treated SCs, suggesting that TTP may play a role in p38α/β MAPK-mediated MyoD regulation.

The rapid p38α/β MAPK-dependent phosphorylation (inactivation) of TTP upon SC activation suggests that TTP plays a role in maintaining SC quiescence. Furthermore, TTP inactivation preceding MyoD up-regulation is consistent with the hypothesis that TTP actively suppresses pro-myogenic mRNA targets. We therefore transfected myofiber-associated SCs with a constitutively active TTP mutant, TTP$_{S52AS178A}$ (TTP-AA-myc), to determine if direct manipulation of TTP influenced MyoD protein levels. TTP-AA-myc is unresponsive to p38α/β MAPK signaling, binds target transcripts, recruits mRNA decay factors, and reduces target mRNA stability (*Stoecklin et al., 2004*). Myofiber-associated SCs transfected with either a control (pcDNA3.1) or a TTP-AA-myc expression plasmid immediately following explant were fixed and scored for Syndecan-4 and MyoD immunoreactivity after 30 hr in culture. The majority (70%) of control transfected cells were MyoD immunoreactive (*Figure 3F,H*), while only 30% of the TTP-AA-myc transfected cells were MyoD immunoreactive (*Figure 3G,H*), demonstrating that TTP reduces myogenic commitment. Importantly, 15% of SCs were MyoD immunoreactive on a subset of myofibers removed immediately following transfection at 6 hr post-explant (dotted line, *Figure 3H*). Thus, ectopic expression of TTP-AA-myc blocked virtually all cells from inducing MyoD post-transfection (*Figure 3H*).

MyoD mRNA in skeletal muscle cell lines is stabilized by HuR (*van der Giessen et al., 2003*) and induction in SCs is concomitant with the loss of TTP (see *Figure 1*). Moreover, TTP may regulate HuR since a polyadenylation variant of HuR mRNA contains an AU-rich element (*Al-Ahmadi et al., 2009*) and blocking p38α/β MAPK signaling prevents HuR induction (*Figure 2—figure supplement 1C,D*). Therefore, HuR is a potential TTP target that may stabilize MyoD mRNA in activated SCs promoting MyoD induction. If correct, HuR expression should be reduced by the TTP-AA-myc mutant similar or to a greater extent then that observed for MyoD. To test this, SCs on intact myofibers were transfected with either a control or TTP-AA-myc mutant plasmid, fixed (as described above) 30 hr post explant and then stained for Syndecan-4 (to identify SCs) and HuR (*Figure 3—figure supplement 1A,B*). Scoring the percentage of Syndecan-4+ cells that were HuR immunoreactive showed that there were no detectable differences in HuR protein expression in control transfected SCs as compared to TTP-AA-myc transfected SCs (*Figure 3—figure supplement 1C*). Thus, it is unlikely that HuR represents a primary TTP target mRNA in activating SCs and may function to control mRNA stability via an alternative signaling pathway.

## TTP binds the MyoD 3′ UTR and regulates MyoD mRNA stability

TTP does not appear to regulate MyoD mRNA by preventing HuR induction and thus, we asked if TTP directly regulated MyoD transcript stability. Bioinformatic analysis of the MyoD 3′ UTR identified a highly conserved TTP binding site (*Figure 4—figure supplement 1A*). To verify TTP binding to the MyoD 3′ UTR, we inserted the full length MyoD 3′ UTR into a Tet-Off β-Globin reporter (referred to as β-MyoD) and performed RNA-immunoprecipitation assays. HEK293T cells were co-transfected with the ß-MyoD construct and a negative control ß-globin sequence that does not bind TTP (β-GAP), along with wild type TTP or an RNA binding-deficient TTP mutant (F126N) (*Lai et al., 2002*), both containing a flag epitope tag. Following transfection, cell lysates were immunoprecipitated with an anti-flag epitope tag antibody. Immunoprecipitation of wild type flag-TTP efficiently co-immunoprecipitated β-MyoD when compared with the negative control β-globin reporter (β-GAP) (*Figure 4A*, lane 1), whereas immunoprecipitation of the flag-TTP$_{F162N}$ mutant failed to co-immunoprecipitate β-MyoD (*Figure 4A*, lane 2). As an internal positive control, wild type flag-TTP co-immunoprecipitated a known target of TTP, the 3′ UTR sequence of Granulocyte Macrophage Colony Stimulating Factor (GM-CSF), (*Figure 4A*, lane 1) that was not co-immunoprecipitated with the flag-TTP$_{F126N}$ mutant (*Figure 4A*, lane 2).

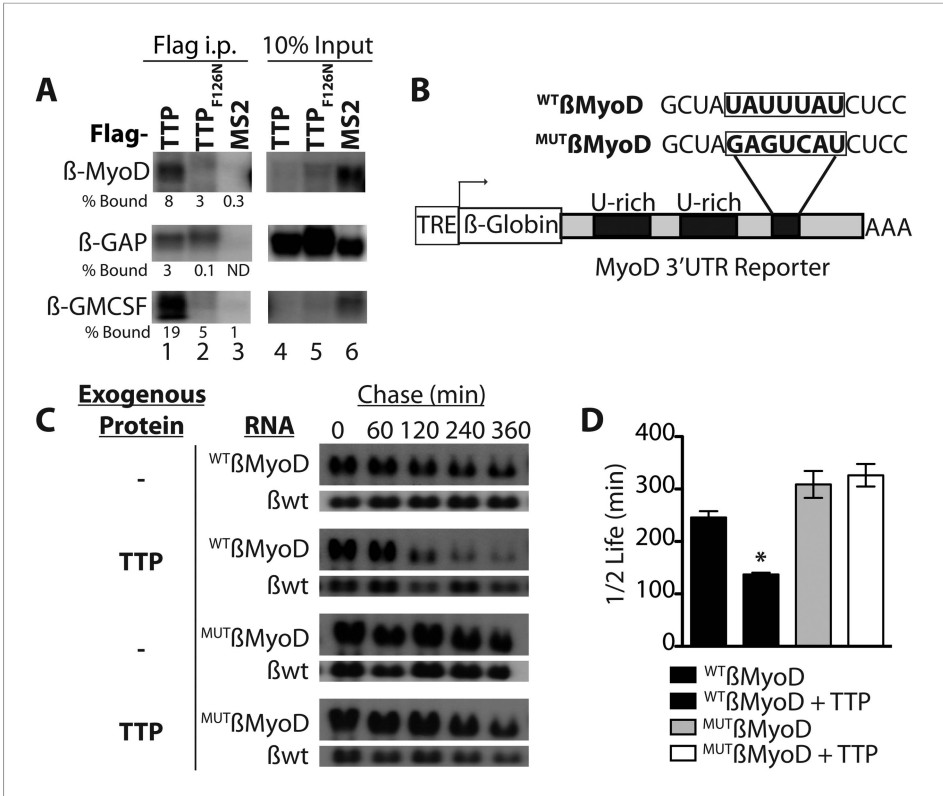

**Figure 4**. TTP binds and regulates the 3′ UTR of MyoD mRNA. (**A**) Northern blots of co-immunoprecipitation assays show TTP binding to the MyoD 3′ UTR in HEK293T cells. Assays were performed with cells co-expressing FLAG-tagged wild type TTP (WT, lanes 1 and 4), an RNA binding mutant of TTP (F126N, lanes 2 and 5), or an unrelated non-RNA binding protein, (MS2, lanes 3 and 6) together with the reporter β-globin mRNA containing the MyoD 3′ UTR (β-MyoD) or the GM-CSF 3′ UTR (β-GM-CSF). Cells were extracted, immunoprecipitated with the indicated flag-tagged protein and associated mRNAs detected by Northern Blot with radioactive anti-sense β-globin probes. A β-globin reporter with no putative TTP binding sites served as an internal negative control (β-GAP). Pellet (lanes 1–3) and 10% input (lanes 4–6) fractions were loaded as indicated. Percentage bound normalized to input RNA indicated (ND-not detectable). (**B**) A schematic of the tetracycline responsive (TRE) β-globin reporter constructs containing the 684 bp MyoD 3′ untranslated region (UTR). Shown are the U-rich HuR binding sites, the putative TTP binding sequence (WTβ-MyoD, bold), and the mutated sequence (MUTβ-MyoD, bold). (**C**) TTP reduces the half-life of a WTβ-MyoD reporter mRNA. Tet-off Hela cells transfected with either pcTET2-WTβ-MyoD or pcTET2-MUTβ-MyoD and CMV-β-Globin, a tetracycline unresponsive loading and transfection control, were subjected to pulse-chase mRNA decay assays and extracts separated by Northern blotting and probed with radioactive anti-sense β-globin probes in the presence of an empty vector control (–) or TTP. (**D**) The calculated β-MyoD mRNA half-life determined from the average ± SEM plotted from at least three independent experiments. Student's two-tailed t-test; * β-MyoD + TTP, p = 0.012.

The following figure supplement is available for figure 4:

**Figure supplement 1**. The MyoD 3′ UTR TTP binding site is conserved and stabilized by constitutive MAPKAPK-2 signaling.

Anti-flag immunoprecipitation of cell extracts expressing an unrelated RNA binding protein, the MS2 viral coat protein, failed to co-immunoprecipitate any of the reporter RNAs (*Figure 4A*, lane 3). These data demonstrate that wild type TTP directly binds the MyoD 3′ UTR.

Active TTP may regulate SC activation by targeting mRNAs for rapid decay, similar to TTP-mediated regulation of inflammation via targeting of proinflammatory mRNAs in macrophages (*Stumpo et al., 2010*). To test whether TTP regulates MyoD mRNA stability, we utilized a Hela cell line stably transfected with the Tetracycline trans-repressor (Tet-Off Hela) to express a β-Globin-MyoD 3′ UTR construct (referred to as β-MyoD) containing a tetracycline responsive element (TRE) upstream of

the promoter (*Figure 4B*). β-MyoD transcription is constitutive until tetracycline addition terminates transcription. At various time points post-chase (following transcription termination), β-MyoD mRNA was quantified and the half-life calculated (*Clement and Lykke-Andersen, 2008*). <sup>WT</sup>ß-MyoD mRNA half-life was 250 min when expressed in Tet-Off Hela cells (*Figure 4C,D*), and co-transfection of a construct expressing TTP destabilized the <sup>WT</sup>ß-MyoD mRNA, decreasing mRNA half-life by nearly twofold (*Figure 4C,D*). Furthermore, TTP-mediated decay of the ß-MyoD mRNA was sensitive to p38α/β MAPK signaling as co-transfection of the TTP construct with a constitutively active MK2 kinase (MK2-EE) increased β-MyoD mRNA stability to ≥350 min (*Figure 4—figure supplement 1B*). MyoD mRNA decay was dependent on the predicted TTP binding sequence as mutation of the putative TTP binding sequence (<sup>MUT</sup>β-MyoD; *Figure 4B*) rendered TTP incapable of destabilizing <sup>MUT</sup>β-MyoD mRNA (*Figure 4C,D*).

## In vivo TTP loss-of-function disrupts muscle homeostasis

TTP may be required for SC quiescence as (1) TTP is present in quiescent SCs, (2) TTP is rapidly inhibited by p38α/β MAPK signaling within minutes of SC activation, and (3) active TTP reduces the half-life of MyoD mRNA. Therefore, loss of TTP from skeletal muscle tissue should precociously activate SCs, promoting inappropriate MyoD expression, and aberrant cell cycle entry. To test this prediction, we analyzed whole tibialis anterior (TA) muscle tissue from uninjured TTP-knockout mice (*Taylor et al., 1996*) and from uninjured TTP/TNFR1/2 triple knockout mice (3KO) (*Carballo et al., 2001*). The triple knockout mice were included to rule out potential muscle phenotypes arising as a secondary consequence of increased inflammation due to systemic stabilization of TNFα mRNA in the absence of TTP (*Stumpo et al., 2010*). Neither the TTP knockout mice nor 3KO mice exhibited any detectable gross morphological differences in muscle sections immunostained with laminin (*Figure 5A*). Moreover, we did not observe Pax7+ cells outside of the basal lamina (*Figure 5—figure supplement 1A*) nor any significant quantitative differences in Pax7+ SCs between wild type and knock out muscle sections when scoring for Pax7 immunoreactivity (*Figure 5—figure supplement 1B*; additional quantification in *Supplementary file 2A*). In contrast, extralaminar and sublaminar Ki67+ cells were increased in TTP knockout mice and 3KO mice, suggesting that SCs exit the quiescent G0 state in the absence of TTP (*Figure 5—figure supplement 1C*). Some sublaminar Ki67+ cells are Pax7 negative, suggesting these cells are myoblasts committed to myogenesis and undergoing terminal differentiation (*Figure 5—figure supplement 1A*). Consistent with increased myogenic commitment of SCs, we observe twofold greater numbers of sublaminar MyoD+ cells in TTP knockout mice and in 3KO muscle compared to wild type mice (*Figure 5A,B*). More centrally located nuclei (CLN) that are indicative of SC fusion into myofibers and a hallmark of muscle injury and repair, were present in TTP knockout mice and 3KO mice compared to controls (*Figure 5C*). Thus, global loss of TTP appears sufficient to drive a low-level injury response in skeletal muscle in the absence of an overt physical or chemical insult.

The relative contributions of non cell autonomous and cell autonomous effects on precocious SC activation cannot be resolved with the global knockout mice and thus, we employed an alternative approach to determine the effects of SC-specific TTP loss-of-function in vivo. We expressed the receptor for RCAS(A) virus, avian leukosis virus receptor (subgroup A), or *tva*, in SCs and infected skeletal muscle tissue with RCAS modified to express U6-driven TTP shRNAs or control shRNAs with a fluorescent CMV-driven mCherry reporter (*Hughes, 2004*; *Harpavat and Cepko, 2006*; *Seidler et al., 2008*; *von Werder et al., 2012*). Since mammalian cells lack *tva* we generated *Pax7<sup>CreERT2</sup>;Rosa26<sup>LSL-tva-lacZ</sup>* mice that express *tva* on Pax7<sup>+</sup> SCs upon tamoxifen-mediated Cre recombinase activation to evaluate RNAi-mediated TTP knockdown in quiescent adult muscle SCs (*Figure 5—figure supplement 2A*).

Tamoxifen-treated *Pax7<sup>CreERT2</sup>;Rosa26<sup>LSL-tva-lacZ</sup>* mice were injected intravenously with RCAS-shRNA-mCherry viruses to infect quiescent SCs (*Figure 5—figure supplement 2B*). Myofibers were not mCherry+ 72 hr post-infection, ruling out direct infection by RCAS viruses (not shown). 3 weeks post infection with a control shRNA (shScramble), we observed mCherry expression in individual cells and in a small percentage of muscle fibers (*Figure 5—figure supplement 2C*, left panel), demonstrating in vivo SC infection and subsequent fusion of infected SCs into myofibers (mCherry+ fibers). A marked increase in mCherry+ fibers was observed in TTP shRNA (shTTPmix) infected muscle compared to muscle infected with scrambled shRNA expressing virus (*Figure 5—figure supplement 2C*, right panel, *Figure 5—figure supplement 2D*).

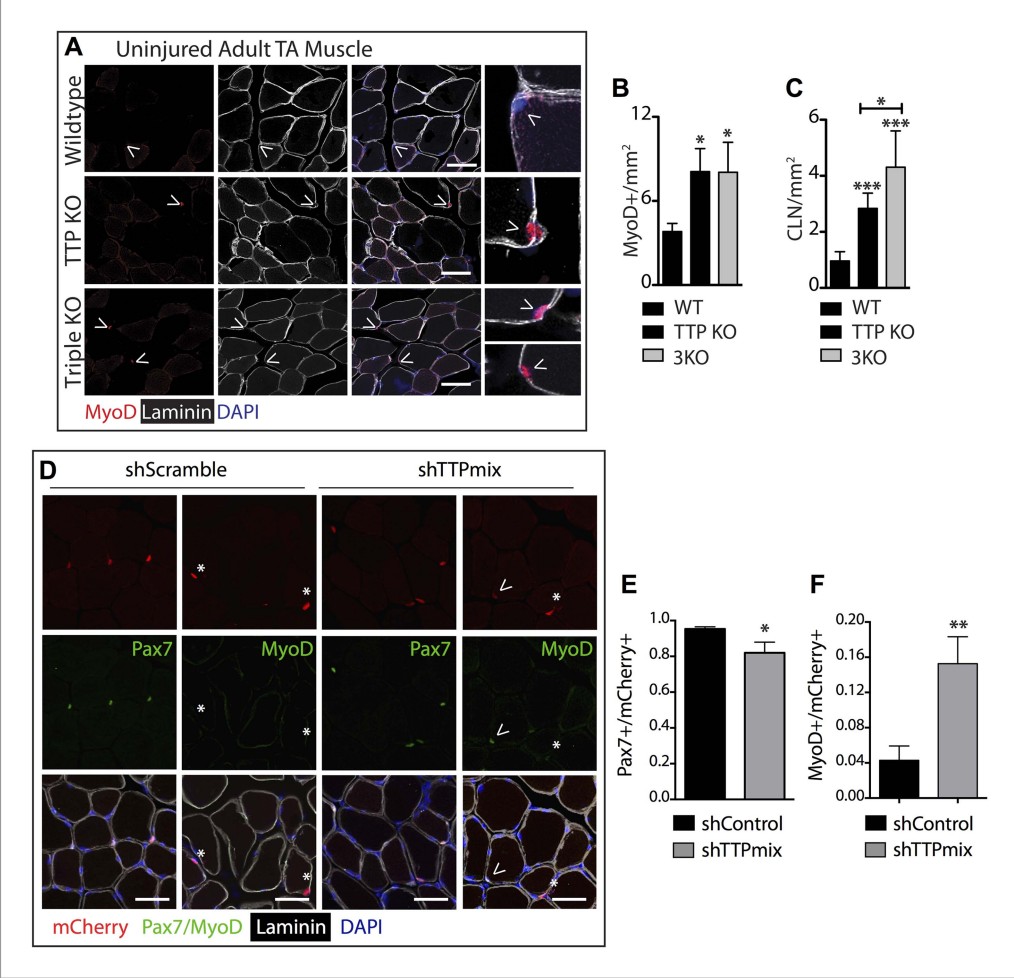

**Figure 5**. TTP loss-of-function promotes promiscuous SC activation. (**A**–**C**) Uninjured TA muscle sections from wild type (WT), TTP knockout, and TTP/TNFR1/2 triple knockout mice (3KO) were stained for MyoD (red) to identify cells committed to myogenesis. Loss of TTP or combined loss of TTP/TNFR1/2 in muscle results in significant increases in MyoD+ cells (**A**, **B**) and centrally located nuclei (quantified in **C**), both hallmarks of actively regenerating muscle. (**D**–**F**) Pax7+ SCs were infected with mCherry-tagged shRNAs and analyzed 3 weeks post-infection to examine the effect of TTP knockdown on SC activation (*Figure 5—figure supplement 2B*). TA muscle cross sections were stained with either Pax7 (green) or MyoD (green) and laminin (white) to determine the activation status of infected (mCherry+) SCs. Shown in (**E**) and (**F**) are bar graphs quantifying the fraction of mCherry+ cells immunoreactive for Pax7 or MyoD, respectively. N = 3 mice for each experimental condition. For all images, nuclei are labeled blue with DAPI and muscle fibers are depicted in white using an anti-laminin antibody; scale bars = 50 µm. In bar graphs, (*) denotes a p-value < 0.05, (**) p < 0.01, (***) p < 0.001 as determined by student's two tailed t-tests.

The following figure supplements are available for figure 5:

**Figure supplement 1**. TTP loss-of-function promotes muscle activation.

**Figure supplement 2**. SC-specific TTP knockdown breaks quiescence in uninjured muscle.

---

Muscle sections from control infected and TTP knockdown infected mice were then visualized for Pax7 and MyoD immunoreactivity and scored to quantify SC activation (*Figure 5D–F*; additional quantification in *Supplementary file 2B*). The mCherry+ cells in control muscles infected with the scrambled shRNA expressing RCAS were >95% Pax7+ with fewer than 5% of the mCherry+ cells immunoreactive for MyoD (*Figure 5D–F*). In contrast, a >threefold increase in MyoD+ immunoreactivity was observed mCherry+ cells in shTTPmix-infected muscle (*Figure 5D,F*). Thus,

SC-specific TTP knockdown breaks SC quiescence, promoting precocious SC activation and enhancing cell fusion with existing myofibers in the absence of muscle injury.

## Discussion

SC homeostasis, the maintenance of quiescence and re-acquisition of quiescence following asymmetric or symmetric cell division is not fully understood. Relatively constant SC numbers are maintained in healthy adult mice prior to and following single or multiple induced muscle injuries (*Cornelison et al., 2004*; *Pisconti et al., 2010*). Elegant experiments have established roles for maintenance of quiescence by microRNAs, where miR-489 represses Dek expression to maintain quiescence, miR-31 represses Myf-5 translation to maintain quiescence, and alternative polyadenylation regulates Pax3 (*Boutet et al., 2012*; *Cheung et al., 2012*; *Crist et al., 2012*). Here, we propose that SC quiescence is actively maintained by mRNA decay, an additional post-transcriptional regulatory mechanism, via the Tis11 family of RNA binding proteins (*Figure 6*). Together, these studies suggest that a host of post-transcriptional regulatory mechanisms maintain the quiescent state of skeletal muscle stem cells and represent an efficient and rapid method for tissue-specific stem cell responses.

Gene chip analysis of SCs isolated from uninjured and injured skeletal muscle and gene chip analysis of quiescent SCs isolated following injection of p38α/β MAPK inhibitors, identified a cohort of transcripts rapidly down-regulated upon SC activation, prior to cell cycle entry (*Farina et al., 2012*). Transcripts encoding proteins that directly regulate mRNAs, including mRNA binding proteins involved in skeletal muscle diseases were highly represented, supporting a critical role for post-transcriptional regulation of RNA in skeletal muscle homeostasis (*Farina et al., 2012*). The involvement of p38α/β MAPK signaling in SC activation (*Jones et al., 2005*), asymmetric SC division (*Troy et al., 2012*) and loss of SC function during aging (*Bernet et al., 2014*) identifies p38α/β MAPK targets as likely mediators of these cellular processes. Therefore, we focused on TTP, which is a p38α/β MAPK signaling target (*Stumpo et al., 2010*) and was down-regulated to the greatest extent during SC activation. The *Tis11/Zfp36* gene family is comprised of TTP and three related genes, of which TTP is the best characterized, containing a tandem zinc finger domain that binds AU rich sequences in the 3′ UTR of mRNAs and targets the mRNAs for sequestration or decay (*Brooks and Blackshear, 2013*).

Potential TTP mRNA targets in SCs would most likely include positive regulators of myogenic commitment. One probable TTP target, HuR, is reported to stabilize MyoD and Myogenin transcripts in myogenic cell lines (*Figueroa et al., 2003*). Although we found no evidence that TTP regulates HuR during SC activation, HuR is rapidly induced upon SC activation, suggesting that critical mediators of myogenic commitment, including the myogenic transcription factors MyoD and Myf5, are regulated by multiple post-transcriptional regulatory pathways. For example, Myf5 is sequestered from translation in quiescent SCs by miR-33 (*Crist et al., 2012*). MyoD is a likely candidate for post-transcriptional regulation as MyoD mRNA is present at low but detectable levels in freshly isolated SCs (*Supplemental file 1C*) (*Cornelison et al., 2001*; *Liu et al., 2013*). MyoD protein is induced within 3h of SC activation (*Jones et al., 2005*) and requires p38α/β MAPK activity (*Jones et al., 2005*). Asymmetric p38α/β MAPK is required for commitment of one daughter cell to myogenesis during asymmetric SC division (*Troy et al., 2012*) and presumably, the TTP binding site in the 3′ UTR of MyoD would permit TTP-mediated mRNA decay in daughter cells lacking activated p38α/β MAPK. The behavior of SCs from aged mice is consistent with p38α/β MAPK-mediated regulation of MyoD as these SCs predominately commit to myogenesis with an accompanying reduction in self-renewal, consistent with their elevated p38α/β MAPK activity (*Bernet et al., 2014*; *Cosgrove et al., 2014*). Although MyoD mRNA appears to be a TTP target, TTP likely regulates other mRNAs with AU-rich sites involved in maintenance of SC quiescence. Furthermore, the Tis11 family members Brf1 and Brf2 were also down-regulated upon SC activation. Ongoing work is focused on identification and characterization of additional TTP/Tis11 targets and their involvement in maintenance of SC quiescence.

In the absence of muscle injury, SC-specific TTP knockdown increased MyoD+ cells, presumably at the expense of Pax7+ progenitor cells. The increase in MyoD+ cells was accompanied by an increase in mCherry+ myofibers when compared to controls where SCs were infected with scrambled shRNAs. Thus, TTP inhibition breaks SC quiescence in uninjured muscle, providing strong evidence for the involvement of TTP in regulating SC homeostasis. Similarly, post-transcriptional regulation of mRNAs by miR-489 maintains SC quiescence, suggesting Dek and potentially other miR-489 targets regulate SC homeostasis (*Cheung et al., 2012*). As we do not know the extent of functional diversity within the Pax7+ SC population it is possible that TTP loss-of-function and miR-489 preferentially affect

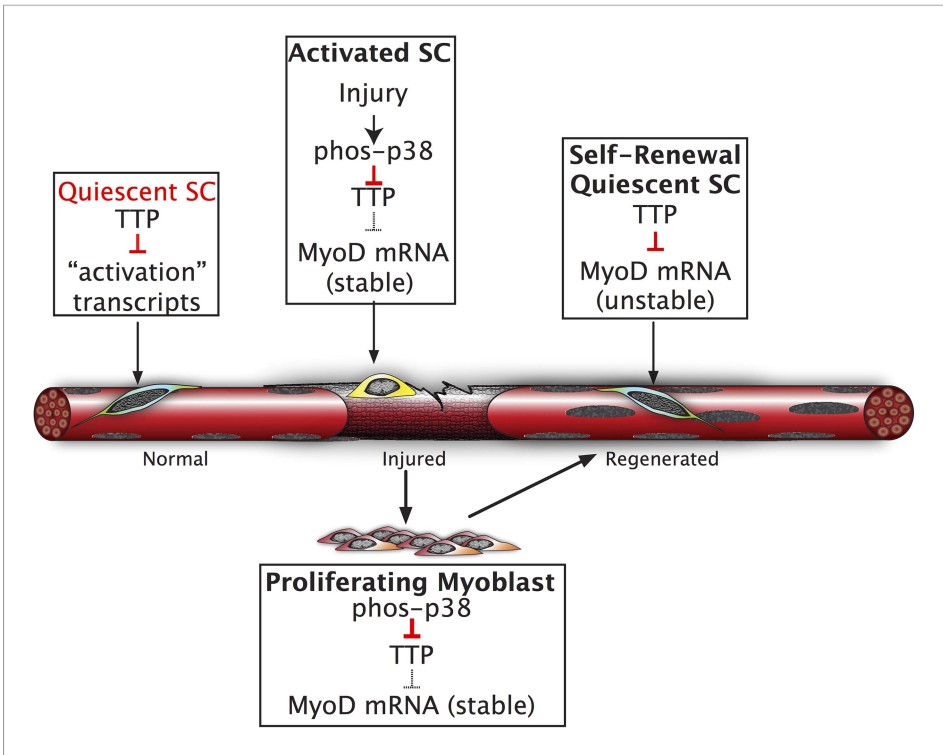

**Figure 6**. Inhibition of TTP by p38α/β MAPK initiates a feed-forward circuit that commits SCs to the myogenic lineage. Muscle regeneration requires SCs to rapidly exit from quiescence but to also renew the quiescent SC population upon completion of regeneration. We propose that TTP suppresses transcripts for SC activation as a mechanism for immediate exit from quiescence. Upon muscle injury, p38α/β MAPK is rapidly activated resulting in phosphorylation of TTP. Phospho-TTP no longer suppresses activation-associated transcripts resulting in rapid induction and SC activation. One such activation-associated transcript targeted by TTP is MyoD, which is rapidly induced due to inactivation of TTP by p38α/β MAPK concurrent with increased MyoD mRNA stability mediated by HuR and transcriptional upregulation of the MyoD gene locus. Since MyoD up-regulates its own transcription, muscle injury would trigger a rapid feed-forward loop resulting in MyoD expression driving SCs to commit to myoblasts. Additionally, we propose that TTP may be involved in down regulation of MyoD mRNA and other TTP-target transcripts expressed by myoblasts, during the initial formation of a quiescent SC population or in reacquisition of quiescence following an injury.

sub-populations of Pax7+ SCs. Additional experiments will resolve whether SC quiescence is coordinately regulated by multiple post-transcriptional mechanisms or whether subsets of SCs are uniquely responsive to distinct skeletal muscle repair or maintenance requirements.

SC quiescence is maintained by multiple post-transcriptional regulatory mechanisms including miRNA-mediated inhibition of Myf5 translation (*Crist et al., 2012*) and Dek translation (*Cheung et al., 2012*). Here, we show that in addition to Myf5 and Pax3 (*Boutet et al., 2012*), MyoD mRNA is regulated post-transcriptionally via a distinct regulatory mechanism whereby MyoD mRNA is targeted for rapid decay in quiescent SCs. Post-transcriptional regulatory circuitry is required for SC homeostasis and appears to maintain SCs in a primed state, permitting a rapid response to skeletal muscle injury. Moreover, post-transcriptional gene regulation likely facilitates SC self-renewal by asymmetric division and may regulate SC homeostasis via multiple mechanisms. In other tissues undergoing rapid turnover, similar post-transcriptional regulatory networks may ensure that a primed stem cell pool is available to repair and maintain tissue function.

## Materials and methods

### Mice
Mice were bred and housed according to National Institutes of Health (NIH) guidelines for the ethical treatment of animals in a pathogen-free facility at the University of Colorado

(Wildtype and Syndecan-4 null line) or at the at the National Institutes of Environmental Health Sciences (TTP and TTP;TNFR1/2 null lines). All animal protocols were approved by the local IACUC. Wild-type mice were C57Bl/6xDBA2 (B6D2F1; Jackson Labs, ME, USA); Syndecan-4 null (*Cornelison et al., 2004*), TTP null (*Taylor et al., 1996*), and TTP;TNFR1/2 null (*Carballo et al., 2001*) mice are previously described. Cells, myofibers and TA muscles were harvested from female mice that were 3–6 months old. For TTP null and TTP;TNFR1/2 null analysis, TA muscle tissues were collected from male and female littermates that were 3–6 months old.

## Microarray analysis and QT-PCR analysis

Isolation and analysis of primary SCs by microarray and QT-PCR analysis was as previously described (*Farina et al., 2012*). CEL files from three replicate gene chips were imported directly into Spotfire (TIBCO, MA, USA) and normalized by GCRMA. Using a 99% confidence threshold (p-value ≤ 0.01), probe sets that changed ≥twofold in relative expression between $Sdc4^{-/-}$ uninjured SCs and $Sdc4^{-/-}$ SCs 12 hr post injury were subtracted from probe sets that changed ≥twofold between wild type uninjured SCs and wild type SCs 12 hr post injury. This analysis resulted in the WT-S4 gene list (*Supplemental file 1A,B*). Gene ontology analysis was performed using Ingenuity Pathway Analysis, CA, USA (*Farina et al., 2012*).

## QT-PCR analysis

For in vitro activation, SCs were isolated from five wild type mice, pooled, and cultured for 0 hr (freshly isolated), 2 hr, 4 hr, 8 hr and 12 hr in F12C containing 15% Horse serum and 2 nM FGF-2; data were collected for two independent experiments. Superscript III RT (Invitrogen, CA, USA) was used for reverse transcription of RNA into cDNA. Fast SYBR Green master mix (Applied Biosystems) was used according to manufacturer's instructions to amplify target transcripts using primers spanning exon/exon junctions for *Elavl1* and *Zfp36*, *Zfp36l1* and *Zfp36l2*. 18S rRNA was used as a reference gene and samples were analyzed in triplicate. Primer sequences: *elavl1* Exon1/2; (FOR: 5′ GCTTATTCGGGGATAAAGTAGCAGGA; REV: 5′ TTCACAAAACCGTAGCCCAAG). *Zfp36* Exon1/2; (FOR: 5′ GCCATCTACGAGAGCCTCCA; REV: 5′ CGTGGTCGGATGACAGGTC). *Zfp36l1* Exon1/2; (FOR: 5′ CGAAGTTTTATGCAAGGGTAA; REV: 5′ GCGCTGGGAGTGCTGTAGTT). *Zfp36l2* Exon1/2; (FOR 5′ CGACCACACTTCTGTCACCCT; REV 5′ GGATTTCTCCGTCTTGCACAA).

## RCAS virus generation and infection

RCASBP(A) vectors containing Gateway cloning sites were obtained from Addgene (MA, USA) (*Harpavat and Cepko, 2006*). Control and TTP shRNA constructs (TRC1/1.5, Sigma–Aldrich, MO, USA) were subcloned into RCAS vectors along with the sequence encoding the florescent reporter mCherry protein. RCAS viruses were generated in DF-1 chicken fibroblast cells using standard protocols and were concentrated and stored at −80C until use (*Flanagan-Steet et al., 2000*). For in vivo studies, Tamoxifen-inducible $Pax7^{CreERT2}$ Cre-driver mice (provided by Dr Gabrielle Kardon) were crossed with a conditional *Tva* receptor line ($ROSA26^{LSL-tva-lacZ}$) (*Seidler et al., 2008*) to generate mice that upon tamoxifen administration express the *Tva* viral receptor on Pax7+ SCs. 3 days following a standard tamoxifen regimen, concentrated RCAS virus was injected (on 3 consecutive days) via tail vein into recipient mice to introduce shRNA and reporter gene constructs into uninjured Pax7+ SCs. 3 weeks post infection, TA muscle was harvested and processed/analyzed as described for immunofluorescence.

## Immunofluorescence of fixed sections

All mice were housed in a pathogen-free facility, and the Institutional Animal Care and Use Committee at the University of Colorado approved all protocols. Whole TA muscles were either perfused or fixed for 2 hr on ice with 4% Paraformaldehyde following dissection from the limb. Muscles were sunk in 30% sucrose and mounted in Tissue Tek O.C.T. mounting media. Cryosections were post-fixed onto slides for 5 min in 4% PFA, permeablized for 5 min with 0.5% TritonX-100 in Phosphate Buffered Saline (PBS), and blocked for 1 hr at room temperature with 5% BSA+0.2% Tx-100. Primary antibodies were used at the following dilutions: 1:250 mouse anti-Pax7 (Developmental Hybridoma Bank, Iowa University), 1:500 rabbit anti-MyoD (Santa Cruz [C-20] sc-304, CA, USA), rabbit anti-Ki67 (ab15580, Abcam, MA, USA), 1:300 Rat anti-Laminin (4HB-2, Sigma-Aldrich, MO, USA), 1:1000 anti-TTP (*Cao et al., 2004*), 1:1000 Chicken anti-Syndecan-4, 1:50 anti-Phospho-MAPKAPK-2 (Thr334) (Cat#3041, Cell Signaling). Sections were stained for 2 hr at room temperature or

overnight at 4°, washed with PBS+0.2% Tx-100 and stained with Alexa Fluor secondary antibodies (anti-rabbit 488, anti-chicken 555, anti-Rat 647, and for mouse IgG detection anti-mouse 488) at a dilution of 1:500 for 1 hr at room temperature. Primary antibody concentrations for Duolink PLA were as follows: 1:200 anti-phospho-MAPKAPK-2 (Thr334) (Cat#3041, Cell Signaling, MA, USA), 1:100 anti-CUGBP1 (sc-20003, Santa Cruz, CA, USA), 1:100 anti-HuR (sc-5261, Santa Cruz, CA, USA), and 1:100 anti-TTP (sc-14030, Santa Cruz, CA, USA). PLA was otherwise performed according to the manufacturer's protocol (Olink Biosciences, Sweden). All immunostaining samples were counterstained with 4',6-diamidino-2-phenylindole (DAPI) and sections were mounted with Vectashield (Vector Labs, CA, USA).

## Immunofluorescence of primary SCs

Primary SCs were dried down on gelatin coated-coverslips in 10 μl PBS in a laminar flow hood and fixed with 4% Paraformadehyde for 10 min at room temperature. Syndecan-4 staining for primary SCs was done as described for myofibers. Primary antibodies were used at the following dilutions: 1:500 mouse anti-HuR, provided by Dr Joan Steitz from Yale University, and 1:200 Rabbit anti-phospho-TTP antibody provided by Dr Georg Stoecklin at the University of Heidelberg in Germany. For scoring fluorescence intensity values, the samples were blinded and masks overlapping Syndecan-4 staining were made. The mean intensity for all fluorescence channels was calculated and background staining was subtracted using Slidebook software (Intelligent Imaging Innovations, CO, USA).

## Preparation, transfection, and immunofluorescence of myofibers

Muscle was dissected from hind limbs and digested for 1.5 hr at 37°C in 400U/ml Type I Collagenase (Worthington, NJ, USA). The muscle slurry was placed into tissue culture dishes containing 15% horse serum in F12C media. Myofibers were teased apart from the muscle using pulled glass pipets and placed into fresh media. Fifty myofibers were transferred to six-well plates containing 2 ml F12C with 15% horse serum and 2 nM FGF-2. Fibers were transfected with 2.75 μg of the pcDNA3-TTP-AA-myc plasmid and 0.25 μg eGFP-N1 plasmid with Lipofectamine 2000 (Invitrogen, CA, USA) according to the manufacturer's instructions for 4 hr. Transfection efficiencies ranged from 25% to 29%. The myofibers were then washed with F12C containing 15% horse serum and 2 nM FGF-2 and incubated for a total of 30 hr in culture. Myofibers were washed with PBS and fixed with 4% paraformaldehyde for 10 min, incubated with 10% goat serum for 1 hr at room temperature, then incubated overnight with 1:1000 chicken anti-Syndecan-4 antibody. The myofibers were washed 3 times with PBS and incubated for 1 hr at room temperature with anti-chicken 1:1000 AlexaFluor 555 and were post-fixed. To detect internal epitopes, myofibers were then permeabilized with PBS containing 0.5% Triton X-100. Following a 1 hr block with 10% goat serum, fibers were incubated with primary antibodies (1:50 mouse anti-MyoD or 1:500 mouse anti-HuR) overnight at 4°C. Fibers were then incubated with secondary anti-mouse Alexa Fluor 647 (1:500) antibodies for 1 hr at room temperature before being mounted on slides with Vectashield containing DAPI.

## Intraperitoneal injection inhibition of p38α/β MAPK with SB203580

In vivo inhibition of p38α/β MAPK was performed using a dosage of 15 mg SB203580 (Alexis Biochemicals, NY, USA) per kg of body weight. SB203580 was resuspended in DMSO and diluted using saline (0.9% NaCl). 1 hr prior to muscle tissue harvest, mice were injected intraperitoneally with 10 μl/g of 1.5 mg/ml SB203580 or a corresponding volume of diluted DMSO as a control. Hind limb muscle tissue was dissected from the leg and immediately placed in 25 μM SB203580 or the DMSO carrier as a control. Primary SCs were isolated as previously described (Cornelison et al., 2004). Briefly, muscle tissue was minced using No. 10 scalpels for 5 min and then incubated for 1 hr at 37°C in 400U/ml Type I Collagenase (Worthington, NJ, USA). The muscle slurry was diluted with F12-C media containing 15% horse serum and sequentially passed through 70 μm and 40 μm filters yielding a single cell suspension. SCs in suspension were fixed immediately in 4% paraformaldehyde for 10 min.

## RNA decay assays with the β-globin-MyoD 3′ UTR reporter

The entire 3′ UTR of MyoD was amplified by PCR from C2C12 myoblast cDNA using the following primers: FOR: 5′ GCATCCATGCGGCCGCGGATGGTGTCCCTGGTTCTT; REV 5′ GCAATCATGCGG CCGCGCGTCTTTATTTCCAACACCT. The PCR product was cloned into NotI sites of the Tetracycline

responsive β-Globin reporter construct as previously described (*Lykke-Andersen et al., 2000*). Quikchange from Agilent (CA, USA) was used to mutate the $^{WT}$ß-MyoD construct according to the manufacturer's instructions using the following primers: FOR: 5′ GGGAGCCCCTTGGGCTA GAGTCATCTCCCAGGCATGCT; REV 5′ AGCATGCCTGGGAGATGACTCTAGCCCAAGGGGCTCCC. Tet-off Hela cells were co-transfected in six-well plates with 2.5 µg of either a wild type or mutant pcTET2-β-MyoD plasmid, 0.05 µg of a CMV-β-Globin plasmid, a tetracycline unresponsive loading and transfection control, and either 0.05 µg of a CMV-wtTTP plasmid, 0.05 µg CMV-wtTTP and 0.5 µg Flag-MK2EE or a pcDNA3 control plasmid. Transcription was pulsed for 6 hr by removal of tetracycline from the media. Following addition of 1 µg/ml tetracycline, RNA was harvested at indicated time points using Trizol according to manufacture's instructions. RNA was separated on a 1.2% Agarose Formaldehyde gel, transferred to a Nylon membrane, probed for β-globin, and the half-life was calculated as previously described (*Clement and Lykke-Andersen, 2008*).

## Flag-tagged immunoprecipitations of TTP bound to the MyoD 3′ UTR

HeLa Tet-off cells in 10-cm plates were transiently transfected with 2.0 µg of the mouse TTP expression plasmids, pcDNA3-Flag-mTTP or pcDNA3-Flag-mTTP$_{F126N}$ (previously described [*Lai et al., 2002*] or 2.0 µg of an expression plasmid containing the N-terminus of the MS2 viral coat protein (previously described [*Lykke-Andersen et al., 2001*]) together with 1.0 µg of a pwtGAP3UAC plasmid (previously described [*Clement and Lykke-Andersen, 2008*]), which lacks a TTP binding site and either 8.0 µg of a pcTET2MyoD plasmid or 9.0 µg of a pcTET2 ATG-GMCSF plasmid. Cells were lysed in 800 µl of RNA immunoprecipitation buffer containing 10 mM Tris pH 7.5, 200 mM NaCl, 2 mM EDTA, 0.1% Triton X-100, 1 µg/ml FLAG Peptide (Sigma-Aldrich, MO, USA), and protease inhibitors 0.5 mM PMSF, 2 µg/ml each of aprotinin and leupeptin. Lysates were incubated with 40 µl (bead volume) anti-FLAG M2 agarose (Sigma) at 4°C for 2 hr. Complexes were washed 8 times with Net-2 (50 mM Tris-HCl pH 7.5, 200 mM NaCl, 0.1% Triton X-100), eluted in 1 ml of Trizol reagent (Invitrogen) and stored at −20°C. Pellet and 10% input fractions were separated on a 1.2% agarose formaldehyde gel, transferred to a nylon membrane, and probed for β-Globin (previously described (*Clement and Lykke-Andersen, 2008*)).

## Acknowledgements

This work was supported by NIH Grants AR039467 and AR04996 to BBO and by NIH Grant GM077243 and American Cancer Society RSG GMC111896 to JL-K. We thank G Stoecklin for providing the anti-phospho-TTP antibody, J Steitz for providing the HuR antibody and D Saur for providing the LSL-Tva mice.

## Additional information

### Funding

| Funder | Grant reference | Author |
| --- | --- | --- |
| American Cancer Society | RSG GMC111896 | Jens Lykke-Andersen |
| National Institutes of Health (NIH) | GM077243 | Bradley B Olwin |
| National Institutes of Health (NIH) | AR039467 | Bradley B Olwin |
| National Institutes of Health (NIH) | AR04996 | Bradley B Olwin |

The funders had no role in study design, data collection and interpretation, or the decision to submit the work for publication.

### Author contributions

MAH, Conception and design, Acquisition of data, Analysis and interpretation of data, Drafting or revising the article; JDD, SLC, ABC, MNH, Acquisition of data, Analysis and interpretation of data, Drafting or revising the article; PJB, This author provided knockout mice essential for the experiments and part of the work by Hausburg was performed in this authors' laboratory, Drafting or revising the article, Contributed unpublished essential data or reagents; JL-A, Analysis and interpretation of data, Drafting or revising the article, Contributed unpublished essential data or

reagents; BBO, Conception and design, Analysis and interpretation of data, Drafting or revising the article, Contributed unpublished essential data or reagents

### Ethics

Animal experimentation: This study was performed in strict accordance with the recommendations in the Guide for the Care and Use of Laboratory Animals of the National Institutes of Health. All of the animals were handled according to approved institutional animal care and use committee (IACUC) protocols (#1012.01, #1104.08) of the University of Colorado-Boulder.

## Additional files

### Supplementary files

• Supplementary file 1. (**A**) Transcripts up-regulated in the WT-S4 gene list. Probesets on Affymetrix arrays that increased in expression ≥ twofold from uninjured to 12 hr post-injury in wild type satellite cells that did not change expression in *Sdc4*$^{-/-}$ satellite cells are tabulated and arranged by decreasing fold-change. The fold change represents the increase in transcript abundance between wild type uninjured satellite cells and satellite cells 12 hr post muscle injury and is reported with the associated p-value. (**B**) Transcripts down-regulated in the WT-S4 gene list. Probesets on Affymetrix arrays that decreased in expression ≥ twofold from uninjured to 12 hr post-injury in wild type satellite cells that did not change expression in *Sdc4*$^{-/-}$ satellite cells are tabulated and arranged by decreasing fold-change. The fold change represents the decrease in transcript abundance between wild type uninjured satellite cells and satellite cells 12 hr post muscle injury and is reported with the associated p-value. (**C**) Myogenic mRNA expression in quiescent SCs. Shown are representative myogenic transcript expression levels from a recently published microarray study of adult, quiescent SCs (*Bernet et al., 2014*). MyoD1 is highlighted in red and other myogenic transcripts are provided as references in black. The entire microarray data set can be accessed using the NCBI Gene Expression Omnibus (GEO) and the unique GEO ID 200047104.

• Supplementary file 2. (**A**) Raw data from TTP, 3KO immunofluorescence quantification. Individual raw data tables (Pax7, Ki67, centrally-located nuclei (CLN), and MyoD) with cell counts per section (40×) for each genotype and biological replicate (WT, TTP KO, 3KO) described in *Figure 5A–C*. On the right of each raw data table is a summary table listing averages, standard deviation, and range for each genotype. (**B**) Raw data from RCAS immunofluorescence experiments. A table listing raw data from RCAS infections plotted in *Figure 5D–F*. For each biological replicate (three shControl and three shTTPmix), the mean, standard deviation, standard error, and confidence interval statistics for mCherry fluorescence, Pax7 immunoreactivity, and MyoD immunoractivity (per 20× section) are provided.

### Major datasets

The following previously published datasets were used:

| Author(s) | Year | Dataset title | Dataset ID and/or URL | Database, license, and accessibility information |
|---|---|---|---|---|
| Bernet JD, Doles JD, Hall JK, Kelly Tanaka K, Carter TA, Olwin BB | 2014 | P38 signaling underlies a cell-autonomous loss of stem cell self-renewal in aged muscle | http://www.ncbi.nlm.nih.gov/geo/query/acc.cgi?acc=GSE47104 | Publicly available at the NCBI Gene Expression Omnibus GSE47104. |
| Cornelison DD, Farina NH, Olwin BB | 2012 | Expression data of satellite cells through muscle injury time course | http://www.ncbi.nlm.nih.gov/geo/query/acc.cgi?acc=GSE38870 | Publicly available at the NCBI Gene Expression Omnibus GSE38870. |

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
