## [Decision Letter]

Thank you for sending your work entitled “Post-transcriptional regulation of satellite cell quiescence by TTP-mediated mRNA decay” for consideration at *eLife*. Your article has been favorably evaluated by Janet Rossant (Senior editor) and three reviewers, one of whom is a member of our Board of Reviewing Editors.

The Reviewing editor and the other reviewers discussed their comments before we reached this decision, and the Reviewing editor has assembled the following comments to help you prepare a revised submission.

The comments of the reviewers converge to a considerable extent. We would ask you to address the points raised and to pay particular attention to the question of the role of TTP in satellite cells (using single fiber experiments for example), TTP effects on transcripts and protein levels, TTP, p38 phosphorylation and messenger RNA stabilisation should be clearly established in this context. The discussion in the paper of the role of TTP in quiescence is not very convincing and if retained requires data on MyoD transcription in these cells, although this does not necessarily have to include in vitro stability assays. The full reviews have been included to assist with your revisions:

Reviewer #1:

This manuscript identifies the RNA binding protein Tristetrapolin (TTP or *Zfp36*) as a factor that is up-regulated in quiescent versus activated satellite cells. The authors show that it binds to AU-rich sequences in the 3' UTR of MyoD mRNA, promoting messenger decay. Interestingly they find that MyoD is already transcribed in quiescent satellite cells and propose that accumulation of this myogenic determination factor, which would lead to myogenesis, is prevented by this post-transcriptional mechanism. Phosphorylation of TTP by p38 MAPK prevents its destabilising action and the authors propose that this takes place on muscle injury when satellite cells are activated and enter myogenesis. Mouse mutant analysis demonstrates the role of TTP in maintaining quiescence since in its absence satellite cells are activated. These results provide new insight into the regulation of satellite cell behaviour and are of interest to the wider field of stem cell research as well as to the muscle community.

A number of points need to be addressed by the authors:

1) HuR is another mRNA binding protein which is known to stabilise MyoD mRNA. Unlike TTP it is expressed at a higher level in activated satellite cells. The authors show that TTP does not regulate HuR, but this does not exclude competition between the factors. Does HuR bind to different sites on the MyoD mRNA? Can the factors bind together and what is the consequence in the critical period of transition at the time of satellite cell activation?

2) TTP is down-regulated at the transcriptional level in activated satellite cells. Why do the authors insist on the functional importance of TTP inactivation by phosphorylation in this context and what is the relative timing of these events? A weakness of the TTP phosphorylation data in this paper is that the antibody to phosphorylated TTP does not work well on sections so that evidence on its in vivo significance is largely circumstantial. More experiments on isolated fibres without TTP overexpression would be helpful.

3) The question of maintenance of satellite cell quiescence, versus return to quiescence after activation, is raised in the paper. The mutant TTP phenotype indicates inappropriate activation in the absence of injury, although this could be a failure of acquisition of quiescence by satellite cells in the postnatal period of growth. Experiments with isolated fibres and siRNA would permit analysis of the role of TTP in the maintenance of Pax7-positive cells that do not differentiate but normally reconstitute the quiescent satellite cell pool.

4) TTP targets pre-inflammatory mRNAs in macrophages and down-regulation of TTP activity leads to up-regulation of TNFa receptors. The authors examine triple mutants as well as the simple TTP mutant, showing rescue of the TTP mutant phenotype. The TTP mutation is not targeted to satellite cells and may therefore exert indirect effects on these cells via the inflammatory response which is known to be important for muscle regeneration. The authors should take this into account.

5) Experiments on TTP and mRNA stability are carried out in HeLa cells. It would be more satisfactory to show the effect on MyoD mRNA in satellite cells. Are many other mRNAs in satellite cells affected?

Reviewer #2:

In this manuscript, Hausburg et al. identify the mRNA decay factor Tristetraprolin (TTP) as a mediator of satellite cell (SC) quiescence. While the data supporting TTP as a regulator of MyoD transcript is convincing, the mechanism by which TTP is regulated from SC quiescence to activation needs further clarification.

1) The microarray and qPCR analyses do not appear to correlate with protein expression analysis of TTP. The authors clearly show that *Zfp36* transcript levels are highest in the quiescent state and this rapidly decreases upon SC activation (either following injury or in culture). At the same time, the authors suggest that in quiescent SCs TTP protein is active and will target its own mRNA for rapid decay. The authors use this reasoning to conclude why they only observe TTP protein expression in 50% of SCs. Yet, their microarray and qPCR analysis would suggest that mRNA levels should be high, and thus, protein levels should be readily detectable. To add to the confusion, the authors suggest that p38α/β activation results in TTP phosphorylation and stabilization of the protein. If this is the case, mRNA transcript levels should also be stabilized since the protein will no longer mediate decay of its transcript. Additionally, the protein should be readily detectable in all SCs due to its increased stability. According to Figure 2, however, this is not the case as the number of TTP+/Sdc4+ cells do not increase following injury. The authors need to re-evaluate and clarify their proposed mechanism for how TTP protein and mRNA are regulated from quiescence to activation.

2) The link made in this manuscript between p38α/β activation and TTP regulation is based solely on previously published work. In this study, however, experiments to confirm that this mechanism is active in SCs were not adequate. Inhibiting p38α/β activation by chemical inhibitors does not necessarily demonstrate a direct regulation by p38α/β on TTP. Does MK2 phosphorylate TTP directly in SCs? Proximity ligation assay between phospho-MK2 and TTP would enhance this claim. Specific siRNA knock-down of MK2 in SCs cultured on myofibers demonstrating reduced TTP phosphorylation is also needed.

3) Figure 3 demonstrates immunoreactivity of phospho-TTP on an isolated SC. The authors should also provide staining of phospho-TTP from SCs cultured on myofibers comparing immediately isolated fibers (quiescent SCs) to SCs cultured on myofibers for 30 hours (activated).

4) Given that TTP is a proposed mediator of SC quiescence, it is very surprising that there were no differences in Pax7+ SC numbers in TTP KO and 3KO mice compared to WT. At what age were these mice examined? Would there be a difference in mice of older ages? Are other TTP family members compensating for the loss of TTP, and are mice with KO of multiple TTP family members available?

5) MyoD and Myf5 are known to be differentially regulated in satellite cells. Does TTP target Myf5? This is an important point to address and may highlight the differences in MyoD vs. Myf5 regulation.

Reviewer #3:

The manuscript by Hausberg and colleagues addresses regulation of MyoD by the RNA binding protein, TTP. The study of RNA binding proteins in stem/progenitor cells is topical. Its role in MyoD regulation, a master regulator of myogenesis warrants investigation. Overall, the authors have done an excellent job of carefully analyzing the expression of TTP during satellite cell quiescence and transition to proliferation, and its regulation of MyoD. In addition, the germline knockout mice show a satellite cell phenotype in vivo showing that TTP is functionally relevant. However at present, I am less clear of the role of TTP in SC function/regulation as it pertains to MyoD.

I have two major points that need addressing: the stability and active transcription of MyoD in quiescence cells and the role of TTP as it pertains to p38 and MyoD on satellite cell function.

1) To conclude that TTP directly impacts MyoD stability in quiescent cells it is important to demonstrate that MyoD is actively transcribed in quiescent cells.

As a suggestion, could the authors perform an in-vitro stability assay whereby extract from quiescent vs activated cells is incubated with transcribed labeled 3'UTR MyoD reporter to see if stability of the reporter is differentially affected? Subsequently immunodeplete TTP and see what happens to reporter stability.

Figure 4: Is it possible to use a MyoD mutant lacking the 3' region and flag tag? In combination with the presented data, this would substantiate the authors conclusions that wild type TTP directly binds the MyoD 3' UTR.

2) The authors correctly conclude that TTP regulates SC biology in some fashion. However the manuscript is not clearly written to pin down the role that TTP plays as it pertains to MyoD and p38.

In the Results, subsection headed “A constitutively active mutant TTP blocks MyoD induction”: The authors use MyoD as a negative marker of SC quiescence. MyoD cannot be used in this case considering that TTP regulates MyoD. A different (independent) marker is needed.

The use of germline mutants demonstrates an increased fraction of Pax7+ SCs express the cycling marker, Ki67. An increased number of MyoD+ cells are also reported. However the total number of Pax7 cells is not altered. Finally, the number of central nuclei is increased.

The authors use the presence of MyoD as a mark of commitment. On its own MyoD alone cannot be used to determine commitment versus cycling satellite cells. Are MyoD+ cells expressing Pax7? Please provide data to substantiate conclusions.

Overall, TTPs role in SC biology needs to be solidified. The authors should incorporate in vitro experiments to determine differences in cell cycle entry, expansion, differentiation and the re-establishment of quiescence.

[Editors' note: further revisions were requested prior to acceptance, as described below.]

Thank you for resubmitting your work entitled “Post-transcriptional regulation of satellite cell quiescence by TTP-mediated mRNA decay” for further consideration at *eLife*. Your revised article has been favorably evaluated by Janet Rossant (Senior editor), a Reviewing editor, and two reviewers. The manuscript has been improved but there are a few remaining issues that need to be addressed before acceptance, as outlined below:

1) The proximity ligation assay (Figure 3) was performed in MM14 cells. As the focus of this study is on the maintenance of SC quiescence, this experiment should be performed on isolated SCs or myofiber associated SCs.

2) If the phospho-TTP antibody works for myofiber staining, it should be used to demonstrate an increase in phospho-TTP within SCs during activation (perform a time course). The authors stated in their response that p38α/β is phosphorylated immediately upon isolation and therefore TTP will already be phosphorylated. However, it is worth examining whether or not an increase in phospho-TTP staining can be detected during the course of SC activation.

3) It is unclear which TTP antibody was used for the myofiber experiment in Figure 3. The text in the manuscript and figure legends suggests that non-phospho TTP antibody was used. The labeling within the figure, however, shows that fibers were stained with anti-phospho-TTP. Please clarify which antibody was used.

4) The number of MyoD+ and central nuclei should be expressed per muscle section in addition to the data already presented. This would help the reader to understand TTP disruption and its effects on activation (MyoD) and fusion (central nucleation) compared to uninjured wildtpe muscle (∼0% MyoD+ and central nucleation) and full injury (∼100% MyoD and central nucleation.

---

## [Author Response]

Reviewer #1:

1) HuR is another mRNA binding protein which is known to stabilise MyoD mRNA. Unlike TTP it is expressed at a higher level in activated satellite cells. The authors show that TTP does not regulate HuR, but this does not exclude competition between the factors. Does HuR bind to different sites on the MyoD mRNA? Can the factors bind together and what is the consequence in the critical period of transition at the time of satellite cell activation?

We have not addressed this point directly, however, our manuscript examines the transition of the satellite cell from the quiescent to an activated state. As we cannot detect HuR in quiescent satellite cells and HuR mRNA is induced when TTP declines, it is not likely that both proteins are present early during SC activation. We do show that TTP is not regulating HuR. During SC activation HuR is not present and knockdown or inhibition of TTP in quiescent SCs induces MyoD. Whether HuR is induced upon TTP knockdown is not known. Since our manuscript focuses on TTP, we believe these potentially interesting experiments are not relevant to the current work but we plan to explore the role of HuR and other RNA binding proteins in SC activation and SC self-renewal.

*2) TTP is down-regulated at the transcriptional level in activated satellite cells. Why do the authors insist on the functional importance of TTP inactivation by phosphorylation in this context and what is the relative timing of these events? A weakness of the TTP phosphorylation data in this paper is that the antibody to phosphorylated TTP does not work well on sections so that evidence on its* in vivo *significance is largely circumstantial. More experiments on isolated fibres without TTP overexpression would be helpful*.

We now show with new data in Figure 3 that a short incubation (24h) with a MK2 inhibitor recapitulates the p38 inhibitor, preventing MyoD protein up-regulation. P38α/β MAPK is the only known kinase that phosphorylates MK2.Furthermore, we establish by PLA that TTP and MK2 are present in a complex in myogenic cells, directly confirming the role of MK2 and providing strong support that the p38α/β pathway regulates TTP activity.

*3) The question of maintenance of satellite cell quiescence, versus return to quiescence after activation, is raised in the paper. The mutant TTP phenotype indicates inappropriate activation in the absence of injury, although this could be a failure of acquisition of quiescence by satellite cells in the postnatal period of growth. Experiments with isolated fibres and siRNA would permit analysis of the role of TTP in the maintenance of Pax7-positive cells that do not differentiate but normally reconstitute the quiescent satellite cell pool*.

We attempted to perform longer drug treatments in culture but ultimately decided an in vivo demonstration that inhibition of TTP disrupts SC quiescence would be the most convincing experiment. We utilized a novel retroviral approach to directly demonstrate that knockdown of TTP specifically in SCs breaks SC quiescence, induces MyoD protein accumulation, is accompanied by exit of SCs from G0, and increases myonuclear accretion. These data demonstrate directly the involvement of TTP in maintaining SC quiescence and SC homeostasis. We have not addressed the role of TTP in SC self-renewal as these experiments are not the focus of the current manuscript and are technically very challenging. Accordingly, references to TTP and SC self-renewal in the narrative were removed.

*4) TTP targets pre-inflammatory mRNAs in macrophages and down-regulation of TTP activity leads to up-regulation of TNFa receptors. The authors examine triple mutants as well as the simple TTP mutant, showing rescue of the TTP mutant phenotype. The TTP mutation is not targeted to satellite cells and may therefore exert indirect effects on these cells via the inflammatory response which is known to be important for muscle regeneration. The authors should take this into account*.

Please see comments above to (3).

5) Experiments on TTP and mRNA stability are carried out in HeLa cells. It would be more satisfactory to show the effect on MyoD mRNA in satellite cells. Are many other mRNAs in satellite cells affected?

Myogenic cells express TTP and endogenous TTP will obscure interpretation of the data. We chose the HeLa system as these cells are nearly devoid of TTP and permit us to directly determine the effects of TTP and the p38α/β pathway on MyoD mRNA stability. It would be difficult to perform similar experiments in myogenic cells as the endogenous MyoD and TTP would pose significant technical challenges in experimental design and interpretation.

We are very interested in other mRNAs regulated by TTP and there are undoubtedly other mRNAs that play critical regulatory roles. We are performing unbiased sequencing experiments to identify these and plan to present the data in a subsequent manuscript.

Reviewer #2:

*1) The microarray and qPCR analyses do not appear to correlate with protein expression analysis of TTP. The authors clearly show that* Zfp36 *transcript levels are highest in the quiescent state and this rapidly decreases upon SC activation (either following injury or in culture). At the same time, the authors suggest that in quiescent SCs TTP protein is active and will target its own mRNA for rapid decay. The authors use this reasoning to conclude why they only observe TTP protein expression in 50% of SCs. Yet, their microarray and qPCR analysis would suggest that mRNA levels should be high, and thus, protein levels should be readily detectable. To add to the confusion, the authors suggest that p38α/β activation results in TTP phosphorylation and stabilization of the protein. If this is the case, mRNA transcript levels should also be stabilized since the protein will no longer mediate decay of its transcript. Additionally, the protein should be readily detectable in all SCs due to its increased stability. According to*
Figure 2*, however, this is not the case as the number of TTP+/Sdc4+ cells do not increase following injury. The authors need to re-evaluate and clarify their proposed mechanism for how TTP protein and mRNA are regulated from quiescence to activation*.

The reviewer has made an excellent point. The reviewer assumes that isolated SCs from which the RNA was obtained are quiescent, however, they are not quiescent but activated. We have previously reported that p38α/β MAPK is activated within minutes of injury and confirm those data here with our analysis of TTP phosphorylation. Thus, the high levels of TTP mRNA observed in freshly isolated SCs are likely due to TTP phosphorylation and subsequent TTP mRNA stabilization. Clearly, TTP mRNA is transcriptionally regulated by mechanisms that are not yet known as the mRNA declines following SC activation.

For Figure 2, the observation was made 1h post BaCl_2_, a short time period after SC activation. Whether this is a sufficient time to change TTP levels is unclear but a worthwhile question. However, the ratio of phospho-TTP to TTP should change dramatically. Unfortunately, the available anti-phospho-TTP antibodies do not work in sections, where we have extensively attempted a variety of techniques to improve phospho-TTP detection. Alternatively, SCs isolated and analyzed in culture are 3-5h or more post activation and do show extensive phospho-TTP immunoreactivity in virtually all cells. For the final point, we agree that additional experiments would solidify the pathway. We have performed several additional experiments, particularly with MK2 that support our conclusions (see new data in Figure 3, Figure 2—figure supplement 1 and Figure 3—figure supplement 1) Finally, to better address the role of TTP in satellite cells, we knocked down TTP in quiescent SCs in the absence of muscle injury. When TTP expression is reduced, SCs break quiescence, induce MyoD, and fuse into existing myofibers. These new data are presented in Figure 5 and Figure 5—figure supplement 2 (see comment (3) for Reviewer 1).

*2) The link made in this manuscript between p38α/β activation and TTP regulation is based solely on previously published work. In this study, however, experiments to confirm that this mechanism is active in SCs were not adequate. Inhibiting p38α/β activation by chemical inhibitors does not necessarily demonstrate a direct regulation by p38α/β on TTP. Does MK2 phosphorylate TTP directly in SCs? Proximity ligation assay between phospho-MK2 and TTP would enhance this claim. Specific siRNA knock-down of MK2 in SCs cultured on myofibers demonstrating reduced TTP phosphorylation is also needed*.

We have addressed both of these excellent points. First, we performed PLA for MK2 and demonstrate that MK2 and TTP are in a complex in myogenic cells. Second, we used a chemical MK2 inhibitor to demonstrate that inhibition of MK2 prevents MyoD protein induction, similar to p38α/β inhibition and constitutive TTP activation. See comment (2) to Reviewer 1.

*3)*
Figure 3
*demonstrates immunoreactivity of phospho-TTP on an isolated SC. The authors should also provide staining of phospho-TTP from SCs cultured on myofibers comparing immediately isolated fibers (quiescent SCs) to SCs cultured on myofibers for 30 hours (activated)*.

First, on immediately isolated fibers SCs are not quiescent but activated. Thus, the experiment as suggested will not work as TTP is phosphorylated upon isolation (See ) as most/all SCs on myofibers are phospho- p38α/β+ upon isolation. We have performed additional experiments to address the concern. First, MK2 is not phosphorylated in uninjured muscle (new Figure 2—figure supplement 1) and inhibition of MK2 prevents TTP phosphorylation and MyoD accumulation (new data as Figure 3).

4) Given that TTP is a proposed mediator of SC quiescence, it is very surprising that there were no differences in Pax7+ SC numbers in TTP KO and 3KO mice compared to WT. At what age were these mice examined? Would there be a difference in mice of older ages? Are other TTP family members compensating for the loss of TTP, and are mice with KO of multiple TTP family members available?

We agree with the reviewer and expected to see a reduction in SCs. While the numbers trend toward reduction, they are not significantly different in the knockout mice. With exercise or aging we expect to see a reduction but these experiments were not performed and are tangential to the focus of this manuscript. We performed a knock-down of TTP specifically in SCs and found that that these cells exit quiescence and fuse into myofibers in the absence of injury (new data in Figure 5 and Figure 5—figure supplement 2). For the knockdown, we do observe a significant reduction in Pax7+ cells (See new Figure 5) in agreement with the reviewer’s expectations. Finally, there are no Brf1 or Brf2 cKO mice available. We are, however, interested in functional compensation between family members and are actively pursuing this hypothesis in a separate manuscript.

*5) MyoD and Myf5 are known to be differentially regulated in satellite cells. Does TTP target Myf5? This is an important point to address and may highlight the differences in MyoD vs. Myf5 regulation*.

We did not test Myf5 directly but given the absence of a consensus AU rich binding site, it is unlikely that TTP targets Myf5 directly. Of particular interest are whether the post-transcriptional regulation of MyoD and Myf5 collaborate or are executed in different subpopulations of SCs. Experiments to test this latter possibility are the subject of an ongoing study and will be published in a future manuscript.

Reviewer #3:

*I have two major points that need addressing: the stability and active transcription of MyoD in quiescence cells and the role of TTP as it pertains to p38 and MyoD on satellite cell function*.

*1) To conclude that TTP directly impacts MyoD stability in quiescent cells it is important to demonstrate that MyoD is actively transcribed in quiescent cells*.

*As a suggestion, could the authors perform an in-vitro stability assay whereby extract from quiescent vs activated cells is incubated with transcribed labeled 3'UTR MyoD reporter to see if stability of the reporter is differentially affected? Subsequently immunodeplete TTP and see what happens to reporter stability*.

We show this by microarray analysis, it has been confirmed and published by the Rando lab and the reference is included. Our unpublished RNA-Seq data show MyoD is present but these cells are activated, not quiescent. Presently, a run-on experiment to directly demonstrate MyoD transcription in *quiescent* SCs is not technically possible. Freshly isolated SCs are *not* quiescent and thus, if performed on freshly isolated cells, these experiments could be difficult to interpret. We realize this is not a fully satisfactory answer but do not believe there are better alternatives at this time.

Figure 4*: Is it possible to use a MyoD mutant lacking the 3' region and flag tag? In combination with the presented data, this would substantiate the authors conclusions that wild type TTP directly binds the MyoD 3' UTR*.

The most convincing data are to mutate the putative binding site and demonstrate that the mutation abrogates binding of TTP and abrogates TTP function. These are precisely the experiments we reported. Removal of the entire 3’ UTR would eliminate any potential secondary binding sites that may contribute to function. Thus, we believe the mutant AU-rich sequence provides the most convincing evidence for the involvement of TTP in regulating MyoD stability. Since the suggested experiment would not add to the data presented we elected not to perform these experiments as they are time consuming and require extensive testing and analysis of new constructs.

*2) The authors correctly conclude that TTP regulates SC biology in some fashion. However the manuscript is not clearly written to pin down the role that TTP plays as it pertains to MyoD and p38*.

We have now knocked down TTP in vivo in SCs and demonstrate directly that these SCs, exit quiescence, accumulate MyoD protein and fuse into myofibers. Additional data presented show that MK2 is present in a complex with TTP and that MK2 inhibition yields results similar to p38α/β inhibition and TTP activation. We believe these data further support a role for the p38α/β pathway regulating TTP activity and MyoD mRNA stability.

*In the Results, subsection headed “A constitutively active mutant TTP blocks MyoD induction”: The authors use MyoD as a negative marker of SC quiescence. MyoD cannot be used in this case considering that TTP regulates MyoD. A different (independent) marker is needed*.

We use Pax7 as well, although we are limited here by the ability to identify truly quiescent SCs. There is no marker of quiescence other than absence of phospho-MK2 or phospho-p38α/β MAPK.

*The use of germline mutants demonstrates an increased fraction of Pax7+ SCs express the cycling marker, Ki67. An increased number of MyoD+ cells are also reported. However the total number of Pax7 cells is not altered. Finally, the number of central nuclei is increased*.

The authors use the presence of MyoD as a mark of commitment. On its own MyoD alone cannot be used to determine commitment versus cycling satellite cells. Are MyoD+ cells expressing Pax7? Please provide data to substantiate conclusions.

We are in agreement with the reviewers but due to the number of markers needed to identify the cells, we have no other choices. We performed additional experiments to specifically knockdown TTP in quiescent SCs in the absence of injury and believe these experiments provide strong support for our conclusions. Please see prior reviewer’s comments and new data in Figures 3 and 5 and Figure 2nd and new data in F, Figure 3–Figure s, 2nd new d and Figure 5—figure supplement 2.

*Overall, TTPs role in SC biology needs to be solidified. The authors should incorporate* in vitro *experiments to determine differences in cell cycle entry, expansion, differentiation and the re-establishment of quiescence*.

Because the TTP mRNA is down-regulated following activation, these experiments would introduce TTP as an artifact and make experimental interpretation difficult. We elected to perform a similar experiment in vivo and report those data in the new Figure 5 and Figure 5—figure supplement 2. The role of TTP in self-renewal and re-establishment of quiescence is an important topic we are keenly interested in and plan a future manuscript on this topic. These experiments are technically challenging and not the focus of the current manuscript which has been reworded to better emphasize our focus on maintaining quiescence and SC homeostasis.

[Editors' note: further revisions were requested prior to acceptance, as described below.]

*1) The proximity ligation assay (*Figure 3*) was performed in MM14 cells. As the focus of this study is on the maintenance of SC quiescence, this experiment should be performed on isolated SCs or myofiber associated SCs*.

We performed the proximity ligation assay using primary isolated SCs and the full spectrum of RNABP antibodies described in the prior submission. We show the new data alongside the MM14 results in an updated Figure 3 demonstrating that pMK2::TTP and pMK2::HuR complexes are present in primary SCs and MM14 cells (see updated Figure 3).

*2) If the phospho-TTP antibody works for myofiber staining, it should be used to demonstrate an increase in phospho-TTP within SCs during activation (perform a time course). The authors stated in their response that p38α/β is phosphorylated immediately upon isolation and therefore TTP will already be phosphorylated – however, it is worth examining whether or not an increase in phospho-TTP staining can be detected during the course of SC activation*.

As stated and shown in the prior revision (Figure 2—figure supplement 1) illustrates, phospho-TTP is detectable in isolated SCs immediately following isolation (approximately 1h 20m post-dissection). Single myofiber isolation takes slightly more time (∼1h 50m) and thus may not be able to capture the [presumably] rapid inactivation (phosphorylation) of TTP. Despite this likely outcome, we performed a detailed short-term time course (0,3,6,12,24h) to query the extent of phospho-TTP in myofiber-associated SCs. Importantly, we confirm that TTP is rapidly inactivated in Pax7+ SCs and remains inactive for at least 24h post-isolation (see new Figure 2—figure supplement 2).

*3) It is unclear which TTP antibody was used for the myofiber experiment in*
Figure 3*. The text in the manuscript and figure legends suggests that non-phospho TTP antibody was used. The labeling within the figure, however, shows that fibers were stained with anti-phospho-TTP. Please clarify which antibody was used*.

We thank the reviewers and apologize for our mistake and the confusion. We used the phospho-TTP antibody in Figure 3. The figure labeling was indeed correct and references to these data in the main text and figure legends now agree and read “phospho-TTP”.

*4) The number of MyoD+ and central nuclei should be expressed per muscle section in addition to the data already presented. This would help the reader to understand TTP disruption and its effects on activation (MyoD) and fusion (central nucleation) compared to uninjured wildtpe muscle (∼0% MyoD+ and central nucleation) and full injury (∼100% MyoD and central nucleation*.

We have now included detailed raw data (and associated statistics as appropriate) for all of the quantified graphs presented in Figure 5. These data can be found in [Supplementary-material SD2-data] (TTP/3KO studies) and [Supplementary-material SD2-data] (RCAS/shRNA studies). These show the data from which all of the graphs were derived.